# Oxidative Stress-Induced HMGB1 Translocation in Myenteric Neurons Contributes to Neuropathy in Colitis

**DOI:** 10.3390/biom12121831

**Published:** 2022-12-07

**Authors:** Rhian Stavely, Lauren Sahakian, Rhiannon T. Filippone, Vanesa Stojanovska, Joel C. Bornstein, Samy Sakkal, Kulmira Nurgali

**Affiliations:** 1Institute for Health and Sport, Victoria University, Western Centre for Health, Research and Education, Sunshine Hospital, St Albans, VIC 3021, Australia; 2Department of Pediatric Surgery, Pediatric Surgery Research Laboratories, Massachusetts General Hospital, Harvard Medical School, Boston, MA 02114, USA; 3Department of Medicine Western Health, The University of Melbourne, St Albans, VIC 3021, Australia; 4The Ritchie Centre, Hudson Institute of Medical Research, Monash Health Translation Precinct, Melbourne, VIC 3168, Australia; 5Department of Anatomy and Physiology, The University of Melbourne, Parkville, VIC 3010, Australia; 6Regenerative Medicine and Stem Cell Program, Australian Institute for Musculoskeletal Science (AIMSS), St Albans, VIC 3021, Australia

**Keywords:** enteric neurons, colitis, plexitis, neuroinflammation, HMGB1, inflammatory bowel disease, oxidative stress, neuropathy

## Abstract

High-mobility group box 1 (HMGB1) is a damage-associated molecular pattern released by dying cells to stimulate the immune response. During cell death, HMGB1 is translocated from the nucleus to the cytoplasm and passively released. High levels of secreted HMGB1 are observed in the faeces of inflammatory bowel disease (IBD) patients, indicating its role in IBD pathophysiology and potential as a non-invasive IBD biomarker. HMGB1 is important in regulating neuronal damage in the central nervous system; its pathological activity is intertwined with oxidative stress and inflammation. In this study, HMGB1 expression in the enteric nervous system and its relevance to intestinal neuroinflammation is explored in organotypic cultures of the myenteric plexus exposed to oxidative stimuli and in *Winnie* mice with spontaneous chronic colitis. Oxidative stimuli induced cytoplasmic translocation of HMGB1 in myenteric neurons in organotypic preparations. HMGB1 translocation correlated with enteric neuronal loss and oxidative stress in the myenteric ganglia of *Winnie* mice. Inhibition of HMGB1 by glycyrrhizic acid ameliorated HMGB1 translocation and myenteric neuronal loss in *Winnie* mice. These data highlight modulation of HMGB1 signalling as a therapeutic strategy to reduce the consequences of enteric neuroinflammation in colitis, warranting the exploration of therapeutics acting on the HMGB1 pathway as an adjunct treatment with current anti-inflammatory agents.

## 1. Introduction

High-mobility group box 1 (HMGB1) is a highly conserved nuclear protein that binds DNA to stabilise the nucleosome and regulate transcription in physiological conditions [1,2]. HMGB1 has emerged as an important alarmin to cellular stress involved in the pathophysiology of several diseases [3]. In conditions of cellular stress, the two HMGB1 nuclear localization sites are hyperacetylated, and HMGB1 translocates to the cellular cytoplasm, where it is released actively by leukocytes or passively by necrotic and necroptotic cells [4]. In the extracellular space, HMGB1 functions as a damage-associated molecular pattern (DAMP), acting as a direct chemoattractant via the C-X-C chemokine ligand 12 (CXCL12) and C-X-C chemokine receptor 4 (CXCR4) axis, as well as stimulating the release of pro-inflammatory mediators via toll-like receptors (TLR)-2 and TLR-4 expressed on nearby immune cells [2]. 

The role of HMGB1 in mediating the immune response has been established in several inflammatory and autoimmune diseases [5,6]. Likewise, high levels of HMGB1 are observed in animal models of experimental colitis and inflammatory bowel disease (IBD) patients [7,8,9,10,11]. The link between HMGB1 and intestinal inflammation is validated by its accuracy as a non-invasive biomarker for paediatric and adult cases of IBD [10,11]. Thus, pharmacological inhibition of HMGB1 is considered a potential therapeutic option for treating IBD [6]. Studies in chemically-induced colitis by dextran sodium sulphate (DSS) have demonstrated that inhibition of HMGB1 can be effective as a preventative treatment against intestinal inflammation [8,9]. Nonetheless, the efficacy of HMGB1 inhibition in established chronic intestinal inflammation has not yet been studied. The compound glycyrrhizic acid (GA) is of interest; it is a specific HMGB1 inhibitor that directly binds to the molecule and inhibits its pro-inflammatory and chemoattractant properties [12]. Furthermore, the effects of GA on the vital nuclear binding properties of HMGB1 are weak; therefore, its use has already been approved for some diseases in the clinic [12,13].

Although HMGB1 is ubiquitously expressed by virtually all cells, its role in the pathophysiology of nervous system damage, including ischemia, reperfusion, haemorrhage and physical trauma, has gathered much attention [14,15,16,17]. In the central nervous system (CNS), HMGB1 appears to be translocated and released almost exclusively from neurons, not glia [15,18]. Therefore, HMGB1 signalling may be a specific neuronal mechanism to activate the inflammatory response. Although extracellularly released HMGB1 can activate leukocytes, enteric neurons also express the major receptors for HMGB1, including TLR-2 and TLR-4 [19,20,21]. Activation of TLRs is known to cause neurotoxicity [22,23,24]; thus, local HMGB1 could potentially contribute to neuronal death. While enteric neuronal loss is common in inflammatory conditions, the mechanisms of cell death in the enteric nervous system (ENS) remain unclear [25]. Cytoplasmic accumulation of HMGB1 is a feature of necrotic cell death pathways; conversely, HMGB1 irreversibly binds to nuclear DNA during apoptosis [3,26]. Therefore, investigating HMGB1 expression may also help to elucidate the pathophysiological mechanism of cell death pathways in enteric neurons during chronic inflammation. 

Oxidative stress and changes in the redox status are critical to the alarmin and DAMP activity of HMGB1. While the functions of HMGB1 are dependent on the redox environment, oxidative stress and reactive oxygen species (ROS) also regulate the translocation of HMGB1 to the cytoplasm and its subsequent release [27]. HMGB1 is important in regulating neuronal damage in the CNS, and its pathological activity is intertwined with oxidative stress and inflammation; both are evident in chronic colitis. In this study, we utilize the Winnie mouse model of spontaneous chronic colitis and ex vivo culture models to elucidate the role of HMGB1 in enteric neuroinflammation.

## 2. Materials and Methods

### 2.1. Animals

For organotypic immunohistochemical studies, male C57BL/6 mice aged 14 weeks were obtained from the Animal Resource Centre (Perth, WA, Australia). For in vivo studies, male *Winnie* mice (C57BL/6 background) aged 14 weeks were obtained from Victoria University (Melbourne, VIC, Australia). *Winnie* mice were compared to age-matched male C57BL/6 mice obtained from the Animal Resource Centre (Perth, WA, Australia). All mice had ad libitum access to food and water and were housed in a temperature-controlled environment with a 12-h day/night cycle. Mice were acclimatised for one week at the Western Centre for Health, Research and Education (Melbourne, VIC, Australia). All mice were culled by cervical dislocation, and the distal portion of the colon was collected for subsequent experiments. All animal experiments in this study complied with the guidelines of the Australian Code of Practice for the Care and Use of Animals for Scientific Purposes and were approved by the Victoria University Animal Experimentation Ethics Committee.

### 2.2. Organotypic Culture of Myenteric Ganglia

The distal colon was collected and pinned in silicon-lined Petri dishes containing Hank’s balanced salt solution (Sigma-Aldrich, Sydney, NSW, Australia) to remove the mucosa and submucosa layers before being cut into 1.5 × 1.5 cm sheets. The organotypic sheet preparations were loosely pinned into 24-well cell culture plates modified to contain a silicon elastomer (Sylgard; Dow Corning, MI, USA) that covered the bottom of the wells with a depth of ~5 mm. Preparations were incubated (37 °C, 5% CO_2_) for 24 h in α-MEM supplemented with 100 U/mL penicillin/streptomycin, 1% glutaMAX and 5% (*v*/*v*) foetal bovine serum (FBS) (Gibco^®^, Life Technologies, Melbourne, VIC, Australia, for all, unless stated otherwise). Hyperoxic oxygen tension, and the chemical H_2_O_2_, were utilised as oxidative stimuli in organotypic preparations. A hyperoxic environment (95% O_2_ and 5% CO_2_) was formed using a self-contained modular incubator (Billups-Rothenberg, Inc., San Diego, CA, USA). Organotypic cultures were exposed to H_2_O_2_ diluted in α-MEM at a final concentration of 100 µM. An equal volume of the α-MEM vehicle was applied to control cultures. Lipopolysaccharide (LPS, Sigma-Aldrich) was applied at 20 µg/mL [1]. Organotypic preparations were cultured for 24 h before being fixed overnight at 4 °C in Zamboni’s fixative (2% formaldehyde and 0.2% picric acid). Preparations were subsequently washed in DMSO (Sigma-Aldrich) (3 × 10 min) to permeabilise the tissue and phosphate buffered saline (PBS) (3 × 10 min) to remove DMSO and the fixative for immunohistochemical experiments.

### 2.3. Glycyrrhizic Acid Administration

HMGB1 was inhibited by glycyrrhizic acid (GA) (Sigma-Aldrich) (10 mg/kg), which was dissolved in 2% cremophor (Sigma-Aldrich): 2% EtOH and 96% sterile water and given to mice via i.p. injections twice daily for 14 days with no less than 8 h between injections. The dose of GA used in our study was effective in reducing oxidative stress, inflammation and neuronal apoptosis in a rat model of ischemic brain injury [2]. The vehicle group received 2% cremophor: 2% EtOH, and 96% sterile water via i.p. injections twice daily for 14 days with no less than 8 h between injections. The volumes for all injections were calculated to each animal’s body weight with a maximum of 200 µL per injection. Mice were culled 24 h after the final treatment for tissue collection.

### 2.4. Evaluation of Colitis 

To analyse faecal water content, faecal pellets were collected from mice and weighed immediately to determine their wet weight. Faecal pellets were dried in a fan-forced oven at 60 °C for 24 h to remove all moisture and weighed again to determine their dry weight. Faecal water content was calculated as the difference expressed as a percentage between the wet and dry weight. Colitis was confirmed by a disease activity index (DAI), which included symptoms of chronic diarrhoea (faecal water content: 60–64% = 1, 65–69% = 2, 70–74% = 3, 75–79% = 4, ≥80% = 5), rectal manifestations (bleeding = 1, prolapse = 2), weight loss (weight before treatment to weight after treatment prior to culling: −1 to −4% = 1, −5 to −9% = 2, ≥−10% = 3), and ratios of colon weight:length from the caecum to the anus (0.0110–0.0140 = 1, 0.0141–0.0160 = 2, 0.0161–0.0180 = 3, 0.0181–0.0200 = 4, ≥0.0200 = 5) [3].

### 2.5. Histology

Distal colon tissues were collected for histology in the optimum cutting temperature (OCT) compound (Tissue Tek, CA, USA) and frozen in liquid nitrogen–cooled isopentane and were cryo-sectioned at 15 μm onto glass slides. Standard hematoxylin and eosin staining (H&E). Sections were immersed in PBS (1 min), rinsed in tap water (30 s), then in hematoxylin (Sigma-Aldrich; 3 min), rinsed in tap water, immersed in Scott’s tap water (1 min), followed by eosin (Sigma-Aldrich; 5 min), rinsed in tap water, immersed in 100% ethanol (1 min) and xylene (4 min). Slides were then mounted with DPX mounting media (Sigma-Aldrich #03989). A histological grading system was used to evaluate gross morphological damage using the following parameters: aberrant crypt architecture (score range, 0–3), crypt abscesses (0–3), leukocyte infiltration (0–3), epithelial damage (0–3), and ulceration (0–3); an average of 3 areas of 500 × 500 µm per section were analysed [4,5]. Slides were imaged using an Olympus BX53 microscope (Olympus Imaging, Sydney, NSW, Australia). All slides were coded, and analysis was performed blindly.

### 2.6. Immunohistochemistry

Antibody labelling of fixed longitudinal muscle and myenteric plexus (LMMP) wholemounts and organotypic preparations was performed using the previously described methodology [6,7]. Wholemounts and organotypic preparations were labelled with the primary antibodies chicken anti-microtubule associated protein (MAP)-2 (1:5000; Abcam, Melbourne, VIC, Australia), rabbit anti-HMGB1 (1:2000; Abcam) and rat anti-CD45 (1:200; BioLegend, San Diego, CA, USA). Tissues were washed with PBS (3 × 10 min) and then labelled with the secondary antibodies Alexa Fluor 594 donkey anti-chicken (1:500), Alexa Fluor 647 donkey anti-rabbit (1:500) and Alexa Fluor 488 donkey anti-rat (1:500) (all from Jackson Immunoresearch, West Grove, PA, USA). Tissues were stained with nuclear marker 4′,6-diamidino-2-phenylindole (DAPI) for 2 min at room temperature prior to being washed with PBS (3 × 10 min) and mounted for imaging. Cross sections of the distal colon from in vivo experiments were prepared and immunolabelled using the previously described methodology [7]. Tissue cross sections were labelled with the primary antibodies chicken anti-MAP-2 (1:5000) and rabbit anti-HMGB1 (1:2000) prior to being washed and labelled with the secondary antibodies Alexa Fluor 594 donkey anti-chicken (1:500) and Alexa Fluor 647 donkey anti-rabbit (1:500). Tissues were then stained with DAPI for 2 min at room temperature prior to being washed with PBS (3 × 10 min) and mounted for imaging. For 8-hydroxy-2′-deoxyguanosine (8-OHdG) immunolabelling, an additional procedure to block endogenous mouse immunoglobin (Ig) was incorporated. Briefly, sections were incubated in 10% normal donkey serum (NDS) (Merck Millipore, Sydney, NSW, Australia) and Triton X-100 at room temperature for 1 h before immunolabelling. Sections were then incubated for an additional 1 h at room temperature with the goat unconjugated affinity purified F(ab) fragment anti-mouse IgG (H+L) antibody (1:100; Abcam, Melbourne, VIC, Australia). This additional blocking step was performed to improve labelling with monoclonal mouse primary antibodies and reduce background labelling [8]. Sections were washed with PBS (3 × 10 min) and incubated overnight at 4 °C with the primary antibodies chicken anti-MAP-2 (1:5000) and mouse anti-8-OHdG (1:200; Abcam). Sections were washed as described above and incubated with Alexa Fluor 594 donkey anti-chicken (1:500) and Alexa Fluor 488 donkey anti-mouse (1:500; Jackson Immunoresearch) for 1 h at room temperature before being mounted onto slides with DAKO fluorescence mounting medium (Agilent Technologies, Melbourne, VIC, Australia) for imaging.

### 2.7. MitoSOX Red Fluorescent Staining

The production of O_2_^−^ in the myenteric plexus was assessed in freshly excised colon segments. Tissues were collected in physiological saline (composition in mmol L^−1^: NaCl, 117; NaH_2_PO_4_, 1.2; MgSO_4_, 1.2; CaCl_2_, 2.5; KCl, 4.7; NaHCO_3_, 25; and glucose, 11), which was gently bubbled with carbogen (95% O_2_–5% CO_2_) throughout the entire procedure. Tissues were viewed under a dissection microscope, cut along the mesenteric border and pinned in a silicon-lined Petri dish containing physiological saline. The mucosa, submucosa and circular smooth muscle layers were removed to expose the LMMP. Tissues were incubated for 40 min at 37 °C in physiological saline containing MitoSOX Red (1:1000) (Molecular Probes^®^, Thermofisher, Melbourne, VIC, Australia). Tissues were then washed in physiological saline (3 × 10 min) prior to being fixed in 4% paraformaldehyde overnight at 4 °C. Paraformaldehyde was removed by washing the tissues with PBS (3 × 10 min) before they were mounted onto glass slides with DAKO fluorescence mounting medium (Agilent Technologies, Melbourne, VIC, Australia) and visualised by confocal microscopy.

### 2.8. Imaging and Analysis

An Eclipse Ti confocal laser scanning system (Nikon, Tokyo, Japan) was used to visualise MAP-2, HMGB1, CD45, 8-OHdG immunofluorescence or the MitoSOX red fluorogenic probe. Identical acquisition settings were used between samples in all experiments. Images were collected as ND2 files containing all metadata, including fluorescence signals at all Z levels. Images were visualised using Image J v1.50b open source software (National Institute of Health, Bethesda, MD, USA) [9,10] with the Image J ND2 Reader plugin and were converted into maximum intensity projections in 16-bit TIFF format. All subsequent analysis was performed using Image J. For all analyses, average mean values were calculated from eight individual images per sample.

For HMGB1 and MAP-2 quantification in wholemount LMMPs and organotypic preparations, Z-series images were randomly acquired using the 40X objective at a thickness of 1 μm. The average MAP-2-immunoreactive (IR) neuronal density was calculated within a 0.1 mm^2^ (316.23 µm × 316.23 µm = 100,000 µm^2^) field of view per image as previously described [11]. In the same images, HMGB1 expression in the ganglia was quantified using the cell counter plugin of ImageJ software. HMGB1 was determined to be expressed by MAP-2-IR neurons or other non-neuronal cells. The intracellular location of HMGB1 in these cells was determined as localised in the nucleus, absent in the nucleus or translocated to the cytoplasm with the aid of the nuclear marker DAPI. HMGB1 expression in cells was reported as the number of cells per 0.01 mm^2^ (100 µm × 100 µm = 10,000 µm^2^) of the ganglionated area or expressed as a percentage of total neurons. HMGB1 expression was pseudo-coloured green for all imaging in wholemounts and cross sections for greater visual distinction. 

Leukocytes immunoreactive for CD-45 were quantified in images of wholemount LMMP preparations. Z-series images were randomly acquired using the 40X objective at a thickness of 1 μm. CD-45-IR cells were enumerated within a 0.1 mm^2^ (316.23 µm × 316.23 µm = 100,000 µm^2^) field of view using the cell counter plugin of ImageJ software. Leukocytes were further classified by their location in relation to the MAP-2-IR ganglia (intra-ganglionic, periphery of ganglia, extra-ganglionic). Values were expressed as the average number of CD45-IR cells per 0.1 mm^2^ area from eight images per mouse. 

To measure 8-OHdG adducts in MAP-2-IR myenteric neurons, Z-series images were randomly acquired using the 60X objective at a thickness of 1μm. Eight randomly captured 16-bit images were analysed with a field of view of 0.04 mm^2^ (200 µm × 200 µm = 40,000 µm^2^) per individual image. Regions of interest were set over MAP-2-IR neurons. Images of 8-OHdG immunofluorescence were converted to binary images by thresholding for high-intensity labelling [12]. Regions of interest defining the location of MAP-2-IR neurons were overlayed onto binary 8-OHdG images. Binary particles were then analysed to obtain the percentage area of 8-OHdG immunoreactivity within the area of myenteric neurons.

Mitochondria-derived O_2_^−^ was visualised using the MitoSOX fluorogenic probe in wholemount LMMP preparations as previously described [13]. Eight Z-series images were randomly acquired using the 40X objective at a thickness of 1 μm; 16-bit images with a field of view of 0.1 mm^2^ (316.23 µm × 316.23 µm = 100,000 µm^2^) per individual image were analysed. Regions of interest were set to determine MitoSOX fluorescence within the myenteric ganglia. The fluorescence intensity of O_2_^−^ in the ganglia was measured as the mean grey value (mean fluorescence intensity) of the pixels within the regions of interest. 

### 2.9. Statistical Analysis

Data analysis was performed using GraphPad Prism v7 (GraphPad Software Inc., San Diego, CA, USA). A one-way ANOVA was performed with a post hoc Holm-Sidak test for multiple comparisons. X and Y correlations were determined using a linear regression analysis with *p* values for significant slope relationships recorded. For all analyses, *p* ≤ 0.05 was considered significant. All data were presented as mean ± standard error of the mean (SEM).

## 3. Results

### 3.1. HMGB1 Is an Endogenous TLR Agonist Highly Expressed in the ENS 

Previous studies implicate TLR2/4 in neuronal loss induced by LPS in cultured enteric neurons in vitro [1]; however, the physiological relevance of these signalling pathways is uncertain. LPS-induced neuropathy was replicated in the organotypic culture of the colonic LMMP preparations. LPS caused a significant reduction in neuronal density (16.7 ± 1.8, n = 7, *p* < 0.001) compared to controls (28.0 ± 1.7, n = 6, Figure 1A–A’’). To identify candidate ligands for TLR signalling in enteric neurons, the publicly available String protein interaction database and the mouse brain single-cell transcriptome database, which contains data on enteric neurons, was utilized [14]. For the protein interaction analysis, only proteins that were part of a physical complex and with validated experimental evidence were included (confidence 0.4). TLR2 and TLR4 have strong physical interactions with the endogenous ligand HMGB1, as reported in immune cells [15] (Figure 1B). A survey of the mouse brain database revealed that HMGB1 expression is particularly high in enteric neurons (inhibitory and excitatory) and enteric glial cells (Figure 1C), making this molecule a strong candidate for local TLR signalling in the ENS during inflammation. 

### 3.2. Chronic Intestinal Inflammation Associates with HMGB1 Translocation in Enteric Neurons 

Expression of HMGB1 was assessed in control mice and *Winnie* mice with chronic colitis. In control mice, HMGB1 was expressed in the nuclei of virtually all cells in the LMMP, with strong staining in neurons and glia in the myenteric ganglia (Figure 2A). In mice with chronic colitis, obvious variations in the expression pattern of HMGB1 were observed, with HMGB1 becoming translocated to the cytoplasm or absent in many neurons, which is characteristic of the secretory mechanism of HMGB1 in cells undergoing necroptosis (Figure 2A). In *Winnie* mice, HMGB1 translocation in myenteric neurons was often observed in clusters, as opposed to sporadically throughout the ganglia, which was suggestive of a localised effect (Figure 2A). In the myenteric ganglia, the number of HMGB1-expressing cells was decreased in *Winnie* mice compared to C57BL/6 controls (*p* < 0.05, Figure 2B, Appendix A, n = 5 animals/group). The number of MAP2^+^ neurons expressing HMGB1 in the nuclei was decreased in *Winnie* mice compared to C57BL/6 controls (*p* < 0.05, Figure 2B’, Appendix A, n = 5 animals/group). Loss of nuclear HMGB1 expression appeared to be specific to the neuronal population, with no effects observed in non-neuronal MAP2^−^ cells within the myenteric ganglia in conditions of chronic inflammation (Figure 2B’’, Appendix A, n = 5 animals/group). Nevertheless, these results were potentially affected by alterations in the neuronal density, as previously reported in *Winnie* mice [16]. To account for this, the expression patterns of HMGB1 were further investigated as a percentage of neurons within each sample (Figure 2C,D). The percentage of neurons without nuclear expression of HMGB1 was increased in *Winnie* mice compared to C57BL/6 mice (*p* < 0.05, Figure 2C, Appendix A, n = 5 animals/group). Given that HMGB1 release is a marker of necroptosis [17] and HMGB1 is an active DAMP, the relationship between HMGB1 expression and myenteric neuronal loss was explored by linear regression analysis. A negative correlation was observed between the proportion of neurons without nuclear HMGB1 and the neuronal density of the myenteric ganglia in vivo (F(1,13) = 14.07, *p* < 0.05, Figure 2C’). Similar to these results, the percentage of neurons with HMGB1 translocation into the cytoplasm was increased in *Winnie* mice compared to C57BL/6 mice (*p* < 0.05, Figure 2D, Appendix A, n = 5 animals/group). The percentage of neurons with cytoplasmic HMGB1 expression negatively correlated with neuronal density (F(1,13) = 9.437, *p* < 0.05, Figure 2D’). Notably, neuronal density did not correlate with HMGB1 expression patterns in non-neuronal cells (F(1,13) = 0.06467, *p* = 0.8).

### 3.3. HMGB1 Translocation in Enteric Neurons and Neuropathy Is Mediated by Oxidative Stimuli 

Cytoplasmic HMGB1 translocation and release have previously been associated with oxidative stress in the CNS neurons [18]. Considering HMGB1 translocation can be redox-sensitive, we explored the effects of chronic intestinal inflammation on oxidative stress in the myenteric ganglia (Figure 3). To determine the levels of oxidative stress in chronic inflammation, DNA/RNA adducts were assessed by immunohistochemistry in cross sections of the distal colon by labelling 8-OHdG (Figure 3A–B’’’). Localisation of 8-OHdG to myenteric neurons was determined by measuring the area of 8-OHdG-IR adducts in myenteric neurons labelled by MAP-2. In myenteric neurons, 8-OHdG was greatly elevated in *Winnie* mice (30.6 ± 5.5%, n = 7 animals) compared to C57BL/6 mice (12.7 ± 3.6%, *p* < 0.05, n = 6 animals) (Figure 3C). High levels of O_2_^−^, as indicated by the intensity of fluorescence, were observed in the myenteric ganglia compared to the muscle layer suggesting that the myenteric plexus is the major source of mitochondrial-derived O_2_^.−^ in LMMPs (Figure 3D,D’). In *Winnie* mice, O_2_^−^ levels (mean fluorescence intensity) in the myenteric ganglia (127.5 ± 22.5 arb. units, n = 6 animals) were more than twice the levels of that observed in control C57BL/6 mice (50.4 ± 3.2 arb. units, *p* < 0.05, n = 5 animals, Figure 3E).

To determine the effects of oxidative stress on HMGB1 expression in the myenteric ganglia, ex vivo organotypic cultures were subjected to hyperoxic (95% O_2_ and 5% CO_2_ incubation) (Figure 4A–C’’’’) and chemical (H_2_O_2_ application) insults (Figure 5A–C’’’’). Culturing organotypic preparations under hyperoxic conditions (n = 7 independent cultures) decreased the total number of neurons with HMGB1 expressed inside the nucleus (Figure 4B–B’’’’) and increased the total number of neurons with cytoplasmic HMGB1 translocation (Figure 4C–C’’’’) compared to controls (Figure 4A–A’’’’) (n = 8 independent cultures) (*p* < 0.05, for both) (Figure 4D,D’, Appendix A). Punctate immunoreactivity of HMGB1 in the cytoplasm was observed in neurons with high levels of HMGB1 translocation. A negative correlation was found between the latter and the neuronal density in the myenteric ganglia (F(1,10) = 13.78, *p* < 0.01) with an R^2^ of 0.5794 (Figure 4E, Appendix A). To account for the decrease in neuronal density caused by hyperoxia, the proportions of neurons with HMGB1 expression were also investigated (Figure 4F–F’’). Similar to total counts, the proportion of neurons with HMGB1 expression in the nuclei decreased (*p* < 0.05), and the proportions of neurons without nuclear HMGB1 (*p* < 0.05) or with cytoplasmic translocation increased (*p* < 0.01) (Figure 4F–F’’, Appendix A). Furthermore, a negative correlation was observed between the proportion of neurons with HMGB1 translocation and neuronal density (F(1,10) = 17.5, *p* < 0.01) with an R^2^ of 0.6364 (Figure 4G, Appendix A). No changes were observed in the total number of non-neuronal cells in the myenteric ganglia that expressed HMGB1 in the nucleus or the cytoplasm after hyperoxia (Figure 4H,H’, Appendix A).

Comparable results were observed in organotypic cultures exposed to H_2_O_2_. Application of H_2_O_2_ (n = 9 independent cultures) decreased the total number of neurons with nuclear HMGB1 expression (Figure 5B–B’’’’) and increased the total number of neurons with cytoplasmic HMGB1 translocation (Figure 5C–C’’’’) compared to controls (*p* < 0.05, for both) (Figure 5D,D’, Appendix A). A negative correlation was observed between the number of neurons with cytoplasmic HMBG1 and neuronal density (F(1,13) = 38.12, *p* < 0.0001, R^2^ = 0.7457) (Figure 5E, Appendix A). Similarly, the proportion of neurons with HMGB1 expression in the nucleus decreased (*p* < 0.05) and the proportion of neurons without nuclear HMGB1 (*p* < 0.05) or those with cytoplasmic HMBG1 (*p* < 0.05) increased (Figure 5F–F’’, Appendix A). The proportion of neurons with HMGB1 translocation was negatively correlated to neuronal density (F(1,13) = 26.21, *p* < 0.001) with an R^2^ of 0.6685 (Figure 5G, Appendix A). Similar to organotypic cultures under hyperoxia, no difference was observed in the number of non-neuronal cells with nuclear or translocated HMGB1 in the myenteric ganglia after treatment with H_2_O_2_ (Figure 5H,H’, Appendix A). Together, these data demonstrate that HMGB1 translocation is regulated by oxidative stress in myenteric neurons and highlights the relationship between HMGB1 and neuronal loss under oxidative conditions.

### 3.4. Inhibition of HMGB1 Alleviates Clinical Symptoms and Myenteric Neuropathy Independent of Plexitis in Chronic Colitis 

To determine the role of HMGB1 in chronic colitis, *Winnie* mice were treated with the HMGB1-specific inhibitor glycyrrhizic acid (GA). Colons collected from sham-treated *Winnie* mice were shorter than those from C57BL/6 and GA-treated mice (Figure 6A). Disease activity scores, consisting of prolapse and bleeding, changes in colon morphology, diarrhoea and weight loss, were evaluated in C57BL/6, sham-treated and GA-treated *Winnie* mice (Figure 6B; n = 6 animals/group). The disease activity index was significantly higher in sham-treated *Winnie* mice (9.0 ± 0.4) compared to C57BL/6 mice (0.1 ± 0.1, *p* < 0.0001). A reduction in DAI (7.3 ± 0.6) was observed in *Winnie* mice treated with GA compared to sham-treated *Winnie* mice (*p* < 0.01). Stool consistency was analysed by measuring the water content in faecal pellets collected from mice before culling (Figure 6C; n = 6 animals/group). The faecal water content in sham-treated *Winnie* mice was significantly higher (77.4 ± 2.1%) compared to C57BL/6 control mice (57.3 ± 0.7%, *p* < 0.001). Treatment of *Winnie* mice with GA reduced the faecal water content (66.5 ± 4.0%) compared to sham treatment (*p* < 0.05), although it remained higher than in C57BL/6 control mice (*p* < 0.05). Likewise, histological evaluation of the colon (Figure 6D,E) indicated that GA treatment reduced mucosal hyperplasia (184.4 ± 15.7 µm, n = 4 animals) compared to sham-treated *Winnie* mice (231.2 ± 8.4 µm, n = 4 animals, *p* < 0.05), with values from control C57BL/6 mice (123.8 ± 8.5 µm, n = 5 animals) remaining lower than both groups (*Winnie* + GA, *p* < 0.01 and *Winnie*-sham, *p* < 0.001). Despite these improvements to DAI, faecal water content and crypt hyperplasia, no differences in histological scoring of colitis were observed after GA treatment, with elevated scores for both sham-treated (8.9 ± 0.9, *p* < 0.01) and GA-treated (8.4 ± 1.7, *p* < 0.05) *Winnie* mice compared to controls (1.1 ± 0.3) (Figure 6D,F, n = 5 animals/group). Given the regulatory role of myenteric neurons in colonic motility and secretion leading to diarrhoea, the effects of HMGB1 inhibition on myenteric neurons were further explored [19,20,21].

The effects of HMGB1 inhibition on myenteric neurons were investigated in wholemount LMMP preparations of the distal colon using the neuronal marker MAP2 (Figure 7A–C’’’’). Assessment of the neuronal density within the ganglia (Figure 7D; n = 6 animals/group) revealed a neuronal loss in sham-treated *Winnie* mice (17.0 ± 1.0 neurons/area) compared to C57BL/6 (22.5 ± 0.7 neurons/area, *p* < 0.01). Treatment of *Winnie* mice with GA attenuated the observed loss of neurons (22.7 ± 1.0 neurons/area, *p* < 0.01). This suggests that HMGB1 is directly implicated in neuronal loss in chronic colitis. To verify this, the pattern of HMGB1 expression in the myenteric ganglia was determined (Figure 7A–C’’’’). The total number of cells in the myenteric ganglia with nuclear HMGB1 was increased in GA-treated *Winnie* mice (44.4 ± 1.4 cells/area) compared to sham-treated *Winnie* mice (32.5 ± 2.7 cells/area, *p* < 0.05) (Figure 7E; n = 5 animals/group). No differences were observed between control C57BL/6 mice (38.5 ± 2.6 cells/area) and GA-treated *Winnie* mice. The increase in cells with nuclear HMGB1 expression in GA-treated *Winnie* mice was correlated with changes in the neuronal population. A reduction of neurons with nuclear HMGB1 was observed in *Winnie*-sham mice (12.2 ± 1.3 neurons/area) compared to C57BL/6 mice (20.1 ± 1.1 neurons/area, *p* < 0.001); this reduction was attenuated by GA treatment in *Winnie* mice (22.7 ± 0.8 neurons/area, *p* < 0.001) (Figure 7E; n = 5 animals/group). No differences were observed between the number of non-neuronal cells expressing nuclear HMGB1 between C57BL/6 mice (18.4 ± 1.8 cells/area), sham-treated (20.3 ± 1.6 cells/area) or GA-treated (21.7 ± 0.8 cells/area) *Winnie* mice (Figure 7E; n = 5 animals/group). The percentage of neurons without nuclear HMGB1 was elevated in *Winnie* mice (34.2 ± 7.0%, *p* < 0.01) compared to control C57BL/6 mice (10.8 ± 3.0%). GA treatment greatly reduced the percentage of neurons without nuclear HMGB1 expression in *Winnie* mice (0.8 ± 0.5%, *p* < 0.001) (Figure 7F; n = 5 animals/group). Likewise, the percentage of neurons with translocation of HMGB1 from the nucleus to the cytoplasm was elevated in *Winnie* mice (11.5 ± 4.5%) compared to control C57BL/6 mice (0.5 ± 0.3%, *p* < 0.05); this was ameliorated by GA treatment, where virtually no cytoplasmic HMGB1 translocation was observed in neurons (*p* < 0.05) (Figure 7G; n = 5 animals/group).

The effects of HMGB1 inhibition on plexitis in the myenteric ganglia of *Winnie* mice were evaluated to determine whether the chemotactic properties of HMGB1 on leukocytes could explain its effects on neuronal loss. Leukocytes visualised by anti-CD45 antibody in LMMP wholemount preparations were observed in proximity to the myenteric ganglia using the pan-neuronal marker MAP-2 (Figure 8A–C’’’). Compared to C57BL/6 mice, both sham-treated and GA-treated *Winnie* mice exhibited higher numbers of leukocytes in the intra-ganglionic region (*Winnie*-sham, *p* < 0.001 and *Winnie* + GA, *p* < 0.01), the periphery of the ganglia (*p* < 0.0001, for both), extra-ganglionic region (*p* < 0.0001, for both) and in total (*p* < 0.0001, for both) (Figure 8D, Appendix A; n = 6 animals/group). No differences were observed between sham-treated and GA-treated *Winnie* mice. These results demonstrate that the neuroprotective properties of HMGB1 inhibition were not mediated by the attenuation of plexitis.

## 4. Discussion

In this study, we identify HMGB1 as an important endogenous cytokine acting locally in the myenteric ganglia to contribute to enteric neuroinflammation. HMGB1 was constitutively expressed in the nuclei of myenteric neurons. A high number of HMGB1 translocation events was observed in the myenteric neurons of *Winnie* mice with chronic inflammation. This directly correlated with the degree of neuronal loss and was associated with elevated oxidative stress in the ganglia. Using organotypic cultures, we demonstrate that translocation of HMGB1 in myenteric neurons is a process regulated by the redox environment and can be induced by oxidative insult. In *Winnie* mice and organotypic models of oxidative injury, neurons had a high specificity for HMGB1 translocation in the myenteric ganglia. Inhibition of HMGB1 translocation by GA averted neuronal loss indicates a direct role of HMGB1 in enteric neuropathies downstream of oxidative stress and local inflammation.

HMGB1 has important physiological functions in the nucleus of cells, binding DNA and regulating transcription; however, it is also an alarmin to cellular stress that can be released extracellularly, eliciting an array of responses depending on the biological context [22]. HMGB1 responds to changes in the redox environment and oxidative stress [17]. These processes are evident in cell death which is closely associated with the pattern of HMGB1 expression [17,22,23]. In our study, HMGB1 was either absent in the nucleus or translocated into the cytoplasm of myenteric neurons. The cellular compartmentalisation of HMGB1 is redox-sensitive, and translocation is induced by intracellular ROS [24]. Hyperacetylation of lysines in the two HMGB1 nuclear localisation sites inhibits HMGB1 entry into the nucleus and results in its accumulation into the cytoplasm [25]. This is a feature of necrotic and necroptotic cell death pathways but not apoptosis, where conversely, HMGB1 irreversibly binds to nuclear DNA [17,22]. Considering the negative association between HMGB1 translocation and neuronal density, myenteric neuronal loss in chronic inflammation and oxidative stress is possibly mediated by a necrotic-like pathway of cell death. The strong relationship between HMGB1 expression and neuronal density suggests that HMGB1 can be a marker of myenteric neuropathy. However, HMGB1 has also been demonstrated to contribute directly to the pathophysiological process of cell death in a broad range of cell types, including renal cells, hepatocytes and neurons of the CNS [26,27,28,29]. GA inhibited HMGB1 translocation and dramatically attenuated myenteric neuronal loss, suggesting that HMGB1 also had a direct role in myenteric neuropathy and did not serve simply as a marker of cell death. Notably, GA exhibits very weak radical scavenging properties; therefore, a direct antioxidant effect is unlikely to explain these effects [30]. Rather, GA has been shown to bind directly to extracellularly released HMGB1 and can inhibit its pro-inflammatory and chemoattractant properties [31]. The effect of GA on the intranuclear binding function of HMGB1 is weak, which may explain why inhibition of HMGB1 by this compound has a low chance of causing deleterious effects and is approved for clinical use [31,32].

In our study, GA inhibited the translocation of HMGB1 into the cytoplasm. Similarly, studies in the CNS have observed that GA inhibits HMGB1 translocation and secretion under conditions of cellular stress [27,28,33,34]. However, it is uncertain whether GA inhibited translocation by binding HMGB1 within the cell or by another mechanism. Considering that GA can weakly affect HMGB1 activity in the nucleus, there may be a degree of affinity for GA to bind HMGB1 intracellularly [31]. Alternatively, it has been postulated that released HMGB1 can induce HMGB1 translocation and further release in cells that are receptive to extracellular HMGB1, including neurons [33,35]. Myenteric neurons expressing translocated HMGB1 were often observed in clusters rather than sporadically throughout the ganglia. The proximity of neighbouring myenteric neurons with HMGB1 translocation may support the notion that released HMGB1 induces further translocation. Myenteric neurons express the major receptors for HMGB1, including the receptor for advanced glycation end-products (RAGE), TLR-2 and TLR-4 [36,37,38]. Given that the TLR-2 and 4 agonist LPS were shown to cause myenteric neuropathy, TLRs appear to be highly relevant to the mechanisms of enteric neuroinflammation. Future studies should clarify whether HMGB1 could also act via RAGE in enteric neurons. Notably, activation of TLR-4 is also known to induce HMGB1 translocation [35]. Thus, it could be predicted that TLR-induced HMGB1 translocation and release can drive further neuropathy.

Although HMGB1 is ubiquitously expressed, the CNS has been a focus for the role of HMGB1 in pathophysiological conditions, including ischemia, reperfusion, haemorrhage and physical trauma [27,28,33,34]. In the nervous system, alterations to HMGB1 expression and its release appear to be specific to neurons [18,33]. In in vitro and in vivo models of experimental subarachnoid haemorrhage, HMGB1 translocation was observed primarily in neurons, whilst HMGB1 translocation events were rare in the glia [18]. Similarly, in a model of traumatic brain injury, only MAP-2 positive neurons exhibited HMGB1 translocation, which was not observed in astrocytes or microglia [33]. This suggests that HMGB1 translocation in nervous system pathologies is relatively specific to neurons. In our study, changes in the number of HMGB1 translocation events were observed in neurons and not in other cell types of the myenteric ganglia in conditions of inflammation and oxidative stress. Therefore, like in the CNS, only neuronal HMGB1 appears to be involved in the pathology of damage to the ENS, particularly driving neuronal loss. 

Mechanisms of neuronal death are highly diverse and can be interrelated. These include traditional caspase-dependent apoptosis pathways and various forms of programmed necrosis or necroptosis [39]. In response to oxidative stress and chronic colitis, myenteric neurons lost expression of HMGB1 in the nuclei and HMGB1 was translocated to the cytoplasm. This suggests that myenteric neurons might undergo necrotic or necroptotic cell death in these conditions. This is further supported by the correlation between the degree of HMGB1 translocation and neuronal loss, in addition to the neuroprotective effects of GA treatments. High levels of ROS are observed in necrosis [17,40]. Likewise, high levels of oxidised DNA/RNA adducts, a marker of oxidative stress-induced DNA damage, were observed specifically in myenteric neurons. Previously, it was observed in a model of dinitrobenzene sulfonic acid (DNBS)-induced colitis that cleaved caspase-3 can be present in 1.4% of neurons at 1.5 h after installation [41]. Comparatively, there was a 42% loss in myenteric neurons [41]. This suggests that a small number of myenteric neurons show signs of apoptosis in the acute stages of chemically-induced colitis; however, other caspase 3-independent or non-apoptotic cell death pathways may also be involved.

HMGB1 is well-established in driving inflammatory and autoimmune diseases [42,43]. In IBD, HMGB1 is implicated in the pathophysiology of intestinal inflammation and has become of interest as a useful faecal biomarker of paediatric and adult IBD [44,45]. In animal models of colitis, increased HMGB1 expression in colon tissues has been observed in mouse models of 2,4,6-trinitrobenzenesulfonic acid (TNBS), DSS chemically-induced colitis and interleukin (IL)-10 deficient mice [46,47,48]. In TNBS-induced colitis and IL-10^−/−^ mice, administration of ethyl pyruvate attenuated colitis and reduced HMGB1 expression [46]. Concomitantly, the antioxidant enzyme haem oxygenase 1 (HO-1) was upregulated, which may indicate that inhibition of oxidative stress contributed to the decrease in HMGB1 [46]. In DSS-induced colitis, the efficacy of direct HMGB1 inhibitors has been evaluated. Daily treatment with dipotassium glycyrrhizate reduced HMGB1 release and attenuated colitis [47]. In a similar model, treatment with an anti-HMGB1 antibody before administration of DSS improved histological scores but had modest effects on clinical scores [48]. Similar results were observed using GA in our study, with a significant but modest reduction in disease activity. However, in our study, GA was applied after chronic inflammation was established; thus, GA was not investigated as a preventative treatment like previous studies and may offer higher efficacy if employed before the onset of inflammation. In studies of colitis, HMGB1 release was associated with a pro-inflammatory role. Nevertheless, in our study, GA treatments did not appear to yield an anti-inflammatory effect near the myenteric ganglia, with high numbers of leukocytes observed in all regions proximal to myenteric neurons despite the inhibition of neuronal HMGB1 translocation. This may suggest that plexitis is upstream of HMGB1 translocation in myenteric neurons. 

In *Winnie* mice, GA treatments paralleled the attenuation of diarrhoea and crypt hyperplasia. Considering that secretion and motility are neurally-regulated processes and dysfunction in the ENS is associated with altered stool consistencies in human pathologies, and experimental models [19,20,21], the prevention of diarrhoea could be a result of reversing neuronal damage. Likewise, ablation of the ENS causes epithelial hyperplasia indicating an important role for mucosal maintenance by the ENS [49], which may explain the correlation between the neuroprotective effects of GA and reduction in epithelial hyperplasia in colitis. Taken together, these results indicate that GA prevents enteric neuropathy mediated by HMGB1 translocation and extracellular release, and this property could be responsible for mitigating diarrhoeal symptoms. Nevertheless, the administration of GA appeared to have a limited effect on inflammation. Signs of active inflammation remained high, and levels of myenteric immune cell infiltrate were not attenuated by GA. These data suggest that GA does not inhibit plexitis and acts via downstream inhibition of HMGB1 signalling in enteric neurons, which prevents further HMGB1 translocation and neuropathy in neighbouring cells. However, it should be considered that high levels of ROS and pro-inflammatory cytokines modify the electrochemical properties and promote dysfunction in enteric neurons [50,51,52,53,54,55]. Therefore, despite GA promoting neuronal survival, perturbations in their function are still possible. 

Considering GA had limited efficacy on leukocyte infiltration, treatment efficacy may be increased in combination with other anti-inflammatory agents to target multiple mechanisms of disease. Therapeutics that act upstream of the HMGB1 pathway may also warrant attention. As HMGB1 translocation was found to be driven by oxidative stress, a combination with antioxidant compounds could be explored to suppress enteric neuroinflammation. However, the limitations of antioxidant compounds, such as low bioavailability and difficulty in achieving therapeutic concentrations in vivo, must also be considered [56,57]. Alternatively, downstream targets of the HMGB1 pathway could also offer a therapeutic avenue. For example, the TLR4 antagonist C34 has already demonstrated efficacy in necrotizing enterocolitis [58]. 

## 5. Conclusions

Translocation of the HMGB1 protein in myenteric neurons was associated directly with neuronal loss in chronic colitis. HMGB1 is ubiquitously expressed by most nucleated cells; however, the specificity for HMGB1 translocation in neurons is a feature of CNS pathologies associated with inflammation and oxidative stress. Likewise, our study demonstrates that translocation of HMGB1 within cells of the myenteric ganglia was only increased in neurons in chronic inflammation conditions. The expression pattern of HMGB1 in myenteric neurons suggests that neuronal loss was possibly caused by necroptotic-like mechanisms. High levels of oxidative stress and ROS production are observed in myenteric neurons in chronic colitis. Oxidative stimuli were demonstrated to directly induce HMGB1 translocation in myenteric neurons; therefore, oxidative stress appeared to instigate HMGB1 translocation and neuronal loss. Inhibition of HMGB1 by GA validated its importance as a key regulator of neuronal loss in vivo. GA did not reduce inflammation or plexitis in the myenteric ganglia, which appear to be upstream events of HMGB1 translocation and neuronal loss. Notwithstanding, therapeutic avenues to modulate HMGB1 signalling warrant exploration as adjunct treatments for colitis to suppress the neuroinflammatory component.

## Figures and Tables

**Figure 1 biomolecules-12-01831-f001:**
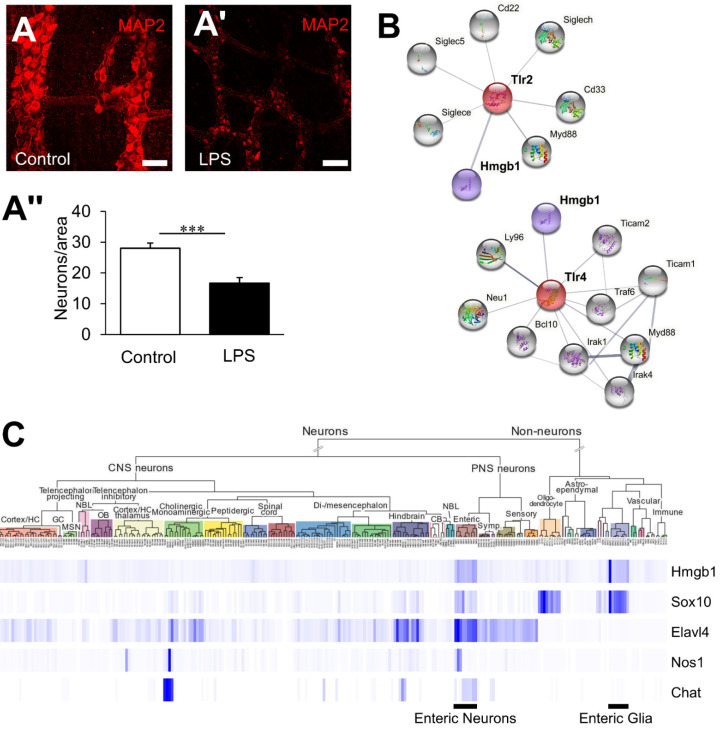
HMGB1 is a candidate endogenous TLR signalling cytokine in enteric neuroinflammation. (**A**,**A’**) Neurons within the myenteric ganglia were observed by immunofluorescence using the neuronal marker MAP-2 in distal colon organotypic cultures. Tissues were cultured for 24 h in standard culture medium (**A**) or in the presence of LPS (scale bar = 50 µm). (**A’’**) Quantification of myenteric neuron density expressed as the number of neurons per ganglionated area. *** *p* < 0.001; control: n = 6 independent samples, LPS: n = 7 independent samples. (**B**) Protein-Protein Interaction network visualisation of proteins with validated experimental evidence of physical interactions with TLR2 and TLR4. (**C**) Gene expression scores of *Hmgb1*; neural-crest and glial cell marker, *Sox10*; neuronal marker, *Elavl4*; inhibitory motor neuron marker, *Nos1*; and excitatory neuron marker, *Chat* from mousebrain.org (accessed on 16 September 2021) [14] single cell database.

**Figure 2 biomolecules-12-01831-f002:**
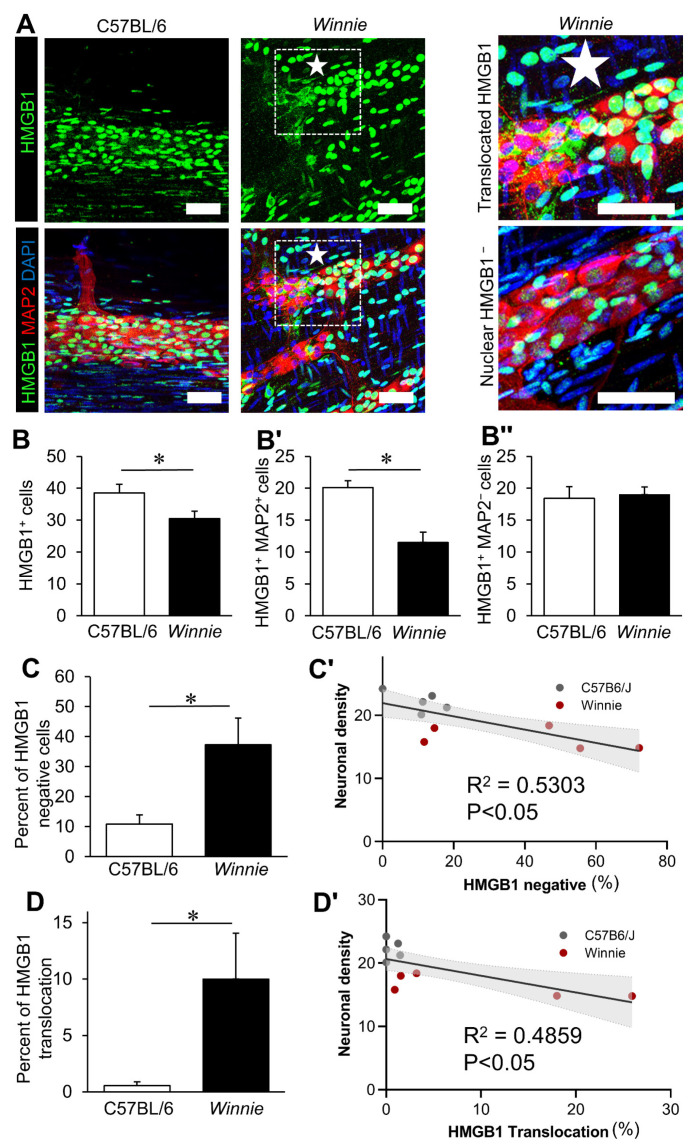
Neuronal translocation of HMGB1 associates with enteric neuropathy in Winnie mice. (**A**) HMGB1 in neurons within the myenteric ganglia was observed by immunofluorescence of HMGB1, the neuronal marker MAP-2 and the nuclear stain DAPI in fresh fixed LMMP wholemount preparations from the distal colon of C57BL/6 and *Winnie* mice. Magnified images demonstrate the unique staining patterns of HMGB1 observed in *Winnie* mice, including the translocation of HMGB1 to the neuronal cytoplasm and loss of nuclear HMGB1 in the neurons but not enteric glia. (scale bar = 50 µm). (**B**–**B’’**) Quantification of the total number of cells (**B**), neurons (**B’**) and non-neuronal cells (**B’’**) with nuclear HMGB1 in the myenteric plexus expressed as cells per ganglionated area. (**C**) Percentage of neurons without HMGB1 in the nucleus. (**C’**) Linear correlation between neuronal density and the percentage of neurons without nuclear HMGB1 expression. (**D**) Percentage of neurons with HMGB1 translocated to the cytoplasm. (**D’**) Linear correlation between neuronal density and the percentage of neurons with cytoplasmic HMGB1 translocation. * *p* < 0.05; C57BL/6: n = 5 animals, *Winnie*: n = 7 animals.

**Figure 3 biomolecules-12-01831-f003:**
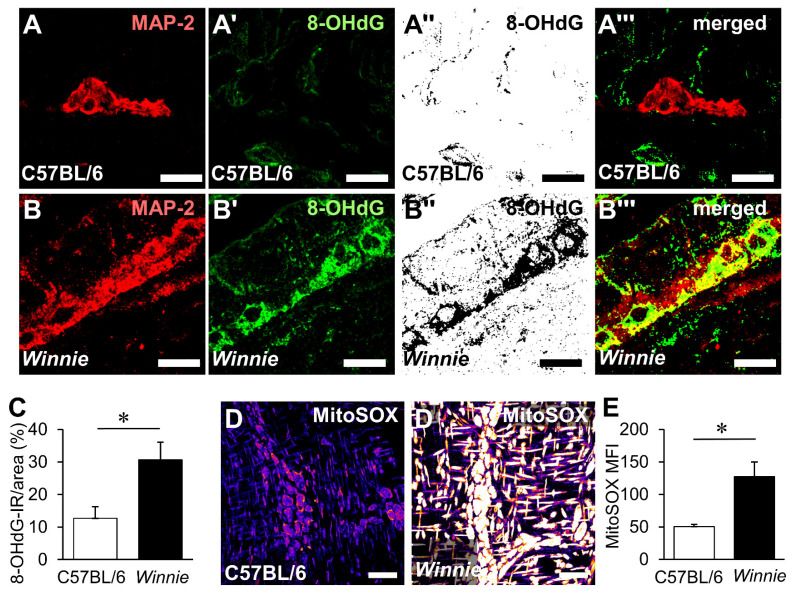
Oxidative stress in the myenteric ganglia of Winnie mice. (**A**–**B’’’**) DNA/RNA oxidative damage in neurons within the myenteric ganglia was visualised by immunofluorescence using the neuronal marker MAP-2 (**A**,**B**) and the DNA/RNA damage marker 8-OHdG (**A’**,**B’**). Adducts of 8-OHdG were observed in binary images of 8-OHdG immunofluorescence (**A’’**,**B’’**). Co-localisation of 8-OHdG adducts with MAP-2 was viewed in merged images in cross sections from the distal colon of C57BL/6 mice (**A**–**A’’’**) and *Winnie* mice (**B**–**B’’’**) (scale bar = 20 µm). (**C**) Quantification of 8-OHdG adducts as a percent of the area in the myenteric neurons in colonic cross sections. * *p* < 0.05; C57BL/6: n = 6 animals, *Winnie*: n = 7 animals. (**D**,**D’**) Mitochondria-derived superoxide (O_2_^−^) in the myenteric ganglia was visualised by the fluorescent probe MitoSOX in LMMP wholemount preparations from the distal colon of C57BL/6 mice (**D**) and *Winnie* mice (**D’**) (scale bar = 50 µm). All images were taken using the same acquisition settings and are pseudo-coloured (LUT: ‘heat’, ImageJ) for greater visual distinction in this figure. (**E**) The mean fluorescence intensity of the myenteric ganglia in single-channel 16-bit images was quantified to determine the intensity of MitoSOX fluorescence. * *p* < 0.05; C57BL/6: n = 5 animals, *Winnie* n = 6 animals.

**Figure 4 biomolecules-12-01831-f004:**
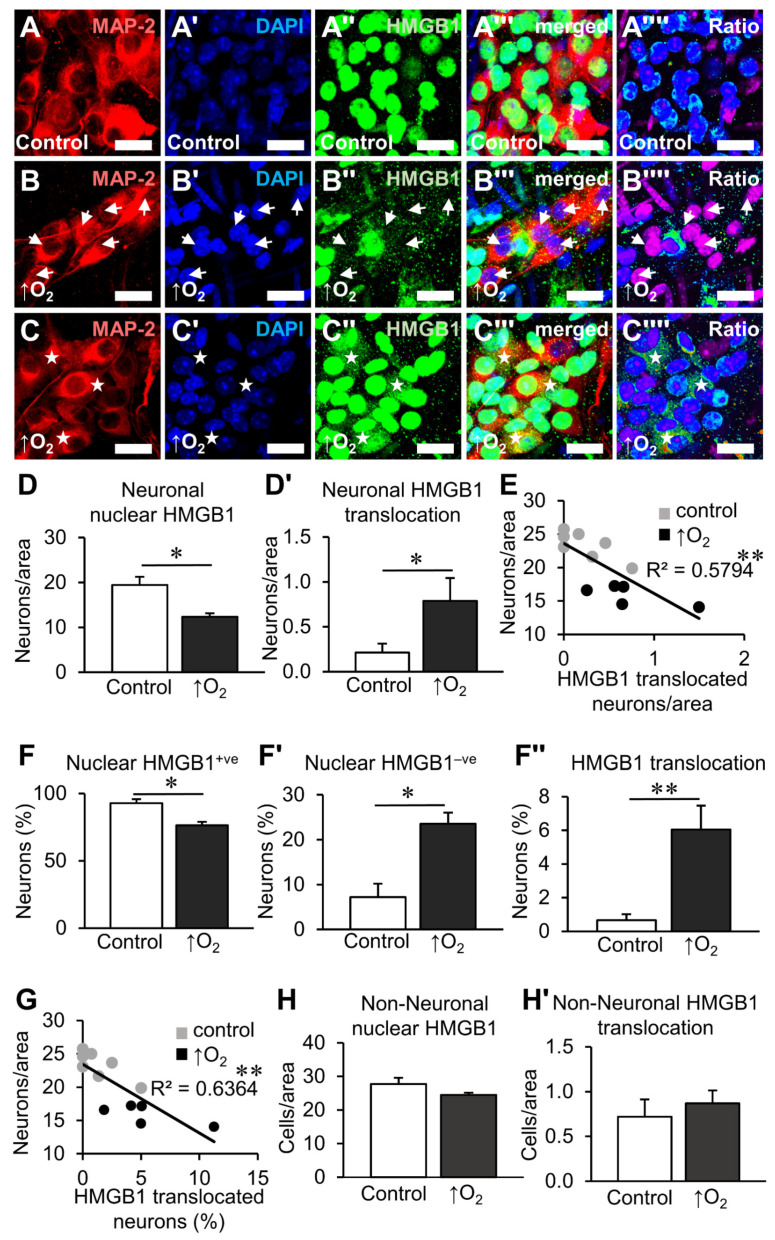
Effects of hyperoxia on HMGB1 expression in myenteric neurons after organotypic culture. (**A**–**C’’’’**) HMGB1 expression within the myenteric ganglia in distal colon organotypic cultures. Myenteric neurons observed by immunofluorescence of the neuronal marker MAP-2 (**A**–**C**), nuclear stain with DAPI (**A’**–**C’**), HMGB1 (**A’’**–**C’’**), merged images (**A’’’**–**C’’’**) and ratio representations of HMGB1:DAPI (**A’’’’**–**C’’’’**) for visual distinction. Tissues were cultured for 24 h in 5% CO_2_ and ambient O_2_ conditions (control) (**A**) or hyperoxic (↑O_2_) conditions (5% CO_2_, 95% O_2_) (**B**,**C**). Neurons without nuclear HMGB1 expression are denoted by arrows (**B**–**B’’’’**), and neurons with HMGB1 translocation into the cytoplasm are denoted by stars (**C**–**C’’’’**) (scale bar = 20 µm). (**D**,**D’**) Quantification of neurons with HMGB1 expressed in the nucleus (**D**) and translocated to the cytoplasm (**D’**) presented as neurons per ganglionated area. (**E**) Linear correlation between neuronal density and neurons with translocation of HMGB1 per area. (**F**–**F’’**) Percentage of neurons with HMGB1 expressed in the nucleus (nuclear HMGB1^+ve^) (**F**), absent in the nucleus (nuclear HMGB1^−ve^) (**F**’) and translocated to the cytoplasm (**F’’**). (**E**) Linear correlation between neuronal density and the percentage of neurons with HMGB1 translocation. (**H**,**H’**) Quantification of non-neuronal cells with HMGB1 expressed in the nucleus (**H**) and translocated to the cytoplasm (**H’**) presented as cells per ganglionated area. * *p* < 0.05, ** *p* < 0.01; control: n = 8 independent samples, ↑O_2_: n = 7 independent samples.

**Figure 5 biomolecules-12-01831-f005:**
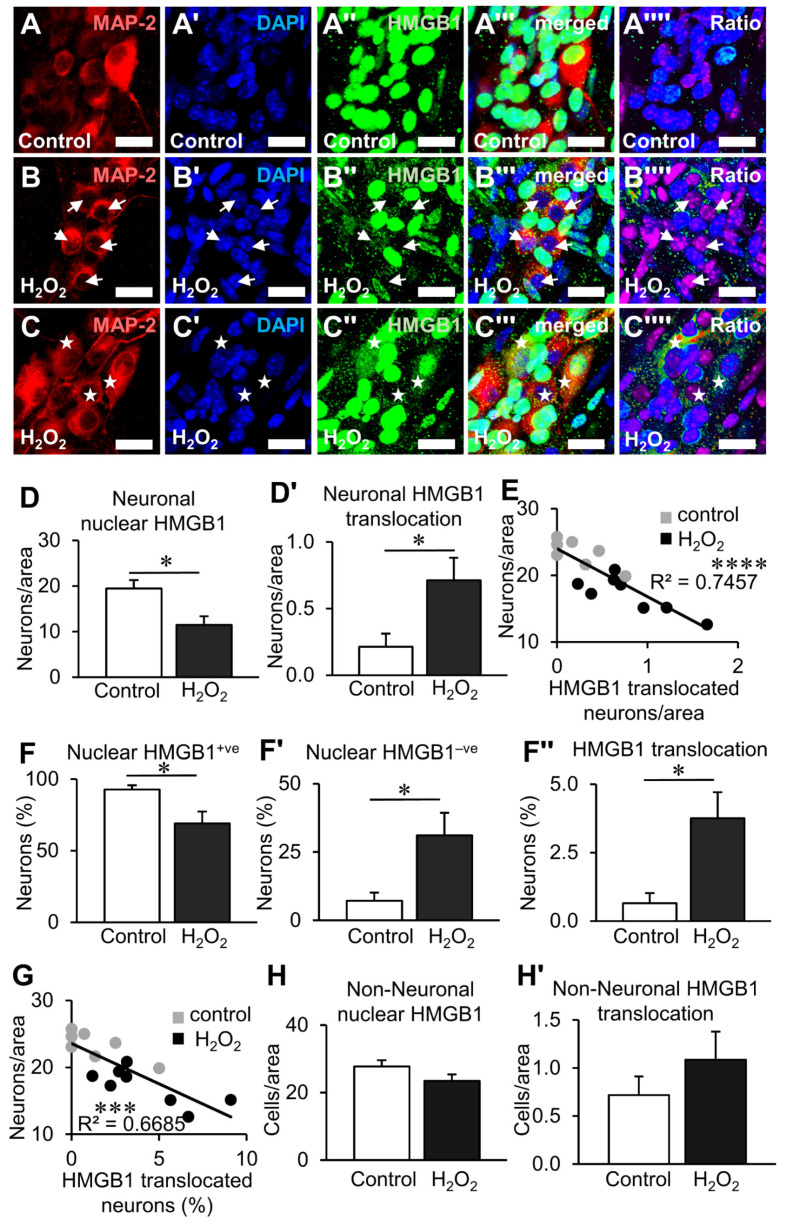
Effects of hydrogen peroxide on HMGB1 expression in myenteric neurons after organotypic culture. (**A**–**C’’’’**) HMGB1 expression within the myenteric ganglia in distal colon organotypic cultures. Myenteric neurons observed by immunofluorescence of the neuronal marker MAP-2 (**A**–**C**), nuclear stain with DAPI (**A’**–**C’**), HMGB1 (**A’’**–**C’’**), merged images (**A’’’**–**C’’’**) and ratio representations of HMGB1:DAPI (**A’’’’**–**C’’’’**) for visual distinction. Tissues were cultured for 24 h in standard culture medium (**A**) or medium with 100 µM H_2_O_2_ (**B**,**C**). Neurons without nuclear HMGB1 expression are denoted by arrows (**B**–**B’’’’**) and neurons with HMGB1 translocation into the cytoplasm are denoted by stars (**C**–**C’’’’**) (scale bar = 20 µm). (**D**,**D’**) Quantification of neurons with HMGB1 expressed in the nucleus (**D**) and translocated to the cytoplasm (**D’**) presented as neurons per ganglionated area. (**E**) Linear correlation between neuronal density and neurons with translocation of HMGB1 per area. (**F**–**F’’**) Percentage of neurons with HMGB1 expressed in the nucleus (**F**), absent in the nucleus (**F’**) and translocated to the cytoplasm (**F’’**). (**E**) Linear correlation between neuronal density and the percentage of neurons with HMGB1translocation. (**H**,**H’**) Quantification of non-neuronal cells with HMGB1 expressed in the nucleus (**H**) and translocated to the cytoplasm (**H’**) presented as cells per ganglionated area. * *p* < 0.05, *** *p* < 0.001, **** *p* < 0.0001; control: n = 8 independent samples, H_2_O_2_: n = 9 independent samples.

**Figure 6 biomolecules-12-01831-f006:**
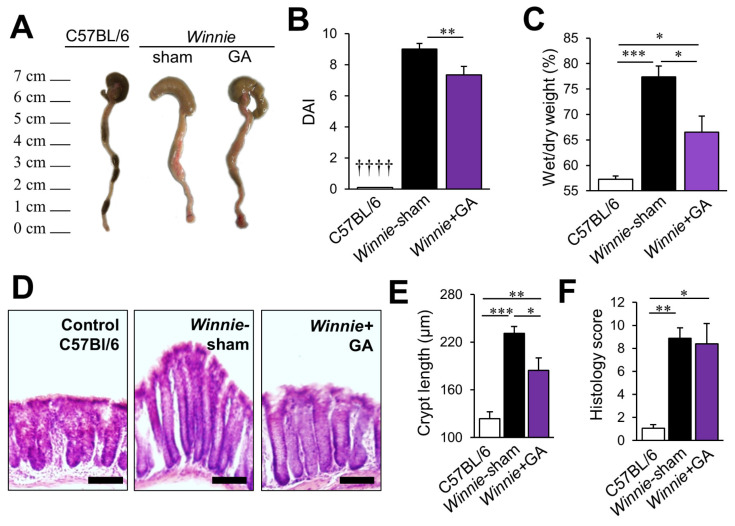
Effects of HMGB1 blocker, glycyrrhizic acid, on colon morphology and the disease activity in Winnie mice. (**A**) Representative photographs of the colons were obtained from C57BL/6 mice, sham-treated *Winnie* mice and *Winnie* mice treated with glycyrrhizic acid (GA). (**B**) Disease Activity Index (DAI) of colitis consisting of rectal bleeding, prolapse, colon morphology, diarrhoea and weight loss in C57BL/6 mice, *Winnie* mice treated with vehicle sham and *Winnie* mice treated with glycyrrhizic acid (GA). ** *p* < 0.01 between *Winnie*-sham and *Winnie* + GA, †††† *p* < 0.0001 between C57BL/6 and both *Winnie*-sham and *Winnie* + GA; n = 6 animals/group. (**C**) Faecal water content was determined by comparing the wet weight to the dry weight of faecal pellets from C57BL/6 mice, *Winnie* mice treated with vehicle sham and *Winnie* mice treated with glycyrrhizic acid. * *p* < 0.05, *** *p* < 0.001; n = 6 animals/group. (**D**) Representative images of haematoxylin and eosin staining (top row) of the cross-sections of the colons from C57BL/6 mice, *Winnie* mice treated with vehicle (sham) and *Winnie* mice treated with glycyrrhizic acid. Scale bars = 100 µm. (**E**) Quantification of the length (µm) of mucosal crypts and in the colon. * *p* < 0.05, ** *p* < 0.01, *** *p* < 0.001; C57BL/6 mice: n = 5; *Winnie* Sham and GA: n = 4 animals/group. (**F**) Quantification of histological grading of colitis in the colon. * *p* < 0.05, ** *p* < 0.01; n = 5 animals/group.

**Figure 7 biomolecules-12-01831-f007:**
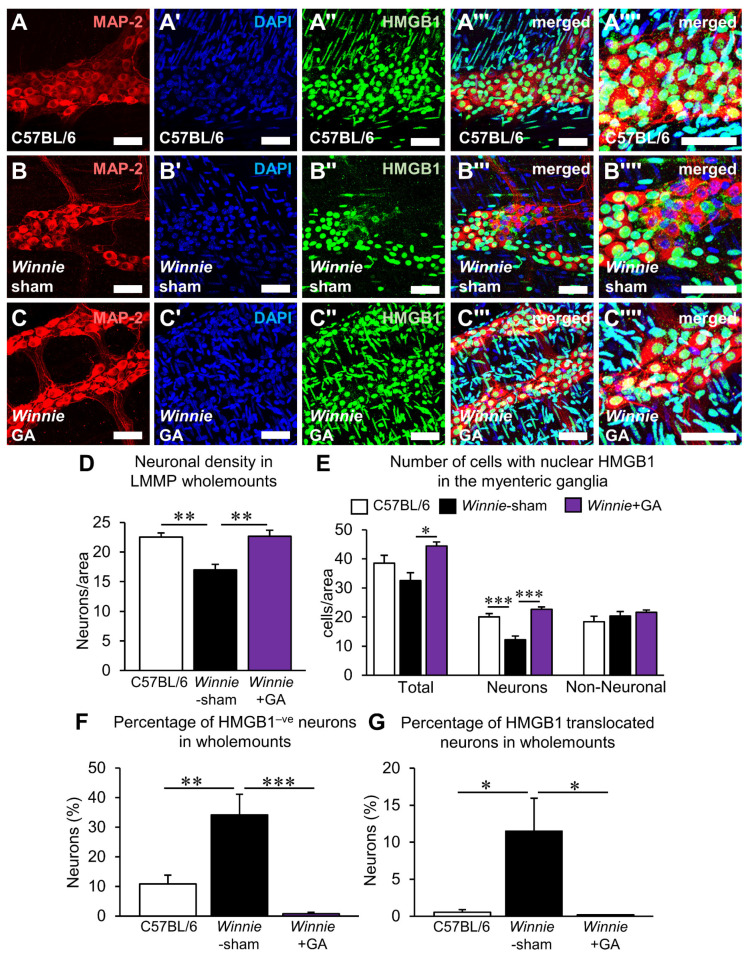
Effects of glycyrrhizic acid treatment on neuronal HMGB1 expression in wholemount LMMP preparations from the distal colon of Winnie mice. (**A**–**C’’’’**) HMGB1 in neurons within the myenteric ganglia were observed by immunofluorescence of the neuronal marker MAP-2 (**A**–**C**), the nuclear stain DAPI (**A’**–**C’**), HMGB1 (**A’’**–**C’’**), merged images (**A’’’**–**C’’’**) and merged magnified images (**A’’’’**–**C’’’’**) in LMMP wholemount preparations from the distal colon of C57BL/6 mice (**A**–**A’’’’**), sham-treated Winnie mice (**B**–**B’’’’**) and Winnie mice treated with glycyrrhizic acid (GA) (**C**–**C’’’’**) (scale bar = 50 µm). (**D**) Quantification of myenteric neuronal density expressed as the number of neurons per ganglionated area. ** *p* < 0.01; n = 6 animals/group. (**E**) The total number of cells, neurons and non-neuronal cells with nuclear HMGB1 within the ganglia expressed as cells per ganglionated area. (**F**) Percentage of neurons without HMGB1 in the nucleus. (**G**) Percentage of neurons with HMGB1 translocated to the cytoplasm. * *p* < 0.05, ** *p* < 0.01, *** *p* < 0.001; n = 5 animals/group.

**Figure 8 biomolecules-12-01831-f008:**
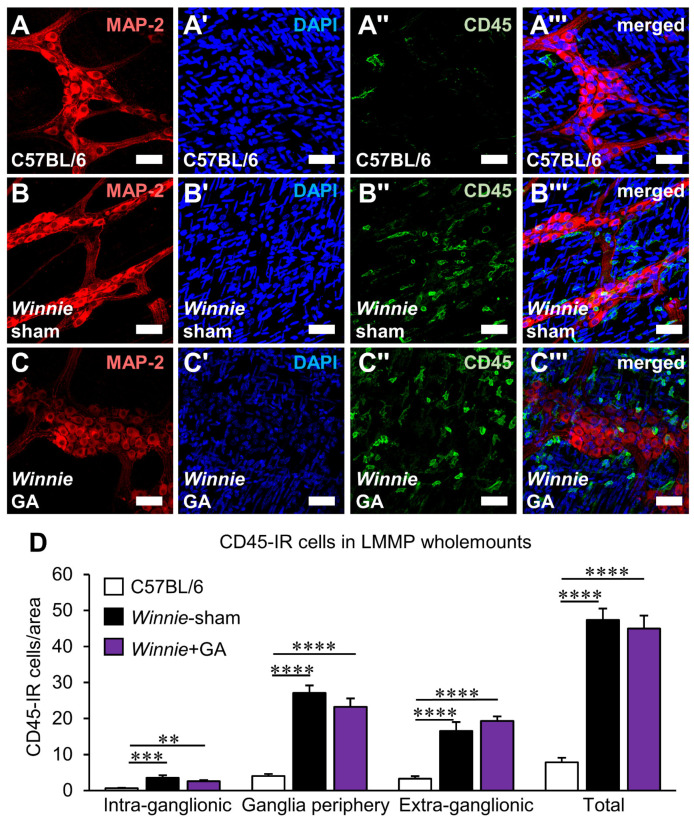
Effects of glycyrrhizic acid treatments on leukocyte numbers in proximity to myenteric neurons in the distal colon of Winnie mice. (**A**–**C’’’**) Leukocytes in proximity to the myenteric ganglia were observed by immunofluorescence of the neuronal marker MAP-2 (**A**–**C**), nuclear stain DAPI (**A’**–**C’**), the pan-leukocyte marker CD-45 (**A’’**–**C’’**) and merged images (**A’’’**–**C’’’**) in LMMP wholemount preparations from the distal colon of C57BL/6 mice (**A**–**A’’’**), sham-treated Winnie mice (**B**–**B’’’**) and Winnie mice treated with glycyrrhizic acid (GA) (**C**–**C’’’**) (scale bar = 50 µm). (**D**) Quantification of CD-45-IR cells per area. Leukocytes were categorised as residing in the intra-ganglionic region, the periphery of the ganglia or the extra-ganglionic region. ** *p* < 0.01, *** *p* < 0.001, **** *p* < 0.0001; n = 6 animals/group.

## Data Availability

All data are presented in the manuscript or made available in the Appendix A. Raw data can be provided upon request to the corresponding author.

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
