# Peer review of "Oxidative Stress-Induced HMGB1 Translocation in Myenteric Neurons Contributes to Neuropathy in Colitis"

_biomolecules, 2022, doi:10.3390/biom12121831_

Round 1
Reviewer 1 Report
This is a fine report on the role of HMGB1 in colitis-driven neuropathy.
There is only a minor concern regarding the manuscript - there is a context of oxidative stress, yet there is no mention whether GA has any component in its activity related to antioxidation. Also, what needs a better discussion, is whether ligands, including GA-like, to HMGB1 are better alone, or should be perhaps co-administered with antioxidants for a full anti-inflammatory and neuron-protective action. Would antioxidants alone be better instead?
Author Response
We are thankful to the Reviewer for their constructive comments.
There is only a minor concern regarding the manuscript - there is a context of oxidative stress, yet there is no mention whether GA has any component in its activity related to antioxidation.
Thank you for this important consideration. Previous studies indicate that GA has a very weak scavenging ability and no direct effects on the scavenging of superoxide or hydroxyl radicals in vitro. For example, GA in the mM range needs to be 20-100 times the concentration to reduce a 1,1-diphenyl-2-picrylhydrazyl (DPPH) radical. The concentrations required for any direct antioxidant effect by GA are not likely to be achieved in vivo (1). The text has been amended on Page 16, lines 560-562.
- Imai, K., Takagi, Y., Iwazaki, A. and Nakanishi, K., 2013. Radical scavenging ability of glycyrrhizin. Free radicals and antioxidants, 3(1), pp.40-42.
Also, what needs a better discussion, is whether ligands, including GA-like, to HMGB1 are better alone, or should be perhaps co-administered with antioxidants for a full anti-inflammatory and neuron-protective action. Would antioxidants alone be better instead?
We agree with the reviewer that the presented data also highlights other potential therapeutic targets that should be explored in the future. This could include therapeutics that act upstream or downstream of HMGB1 release and function. As HMGB1 translocation was found to be driven by oxidative stress, antioxidant compounds could be explored to suppress enteric neuroinflammation. However, the limitations of antioxidant compounds, such as low bioavailability and difficulty in achieving therapeutic concentrations in vivo, must also be considered (1, 2). Alternatively, downstream targets of the HMGB1 pathway could also offer a therapeutic avenue. For example, the TLR4 antagonist C34 has already demonstrated efficacy in necrotizing enterocolitis (3). We have modified the Discussion to highlight these important considerations provided by the reviewer. Page 19, lines 665-674.
- J. A. Baur, D. A. Sinclair, Therapeutic potential of resveratrol: the in vivo evidence. Nature Reviews Drug Discovery 5, 493-506 (2006).
- H. J. Forman, H. Zhang, Targeting oxidative stress in disease: promise and limitations of antioxidant therapy. Nature Reviews Drug Discovery 20, 689-709 (2021).
- M. D. Neal et al., Discovery and Validation of a New Class of Small Molecule Toll-Like Receptor 4 (TLR4) Inhibitors. PloS one 8, e65779 (2013).
Reviewer 2 Report
Authors has interestingly reported novel perspective of oxidative stress induced translocation of HMGB1. Study is great contributing to the current knowledge. I have few suggestion to this study.
1) What is the underlying mechanism behind the effect of GA on clinical symptoms and myenteric neuropathy.
2) Why does author has neglected the aspect of RAGE which is a dominant binding partner of HMGB1..
3) I recommend to include the graphical abstract representing the experimental work flow, which will ease the reader.
Author Response
Authors has interestingly reported novel perspective of oxidative stress induced translocation of HMGB1. Study is great contributing to the current knowledge. I have few suggestion to this study.
We are thankful to the Reviewer for their constructive comments.
1) What is the underlying mechanism behind the effect of GA on clinical symptoms and myenteric neuropathy.
In the present study, our data indicate that oxidative stimuli cause a redox-sensitive translocation and release of HMGB1 in enteric neurons. We speculate that HMGB1 causes further neuronal death in neighbouring enteric neurons after its extracellular release. GA has well-characterised HMGB1 binding properties, which inhibit its activity and was shown to be effective in preventing enteric neuropathy in the presented data. In the same mice, we observe a decrease in diarrhoeal symptoms which may be explained by the neuroprotective effects of GA given the nervous system controls motility and secretion. We have edited the discussion to include this potential mechanism of action explaining our results. Page 18 Line 594, Page 19 Lines 654-656, 660-661.
2) Why does author has neglected the aspect of RAGE which is a dominant binding partner of HMGB1.
We agree with the reviewer that RAGE could also be involved in the mechanism of neuroinflammation mediated by HMGB1. HMGB1 is a known endogenous agonist for RAGE, TLR-2 and TLR-4. Given the TLR-2 and 4 agonist LPS was shown to cause myenteric neuropathy, TLRs appear to be highly relevant to the mechanisms of enteric neuroinflammation. Future studies should clarify whether HMGB1 could also act via RAGE in enteric neurons. Page 18 Line 589-592.
3) I recommend to include the graphical abstract representing the experimental work flow, which will ease the reader.
We greatly appreciate this suggestion and have added a graphical abstract to make interpretation of the study easier.